

# Phase Segmentation of X-Ray Computer Tomography Rock Images using Machine Learning Techniques: an Accuracy and Performance Study

Swarup Chauhan[1], Wolfram Rühaak[1,2], Hauke Anbergen[3], Alen Kabdenov[3], Marcus Freise[3],
Thorsten Wille[3], Ingo Sass[1,2]

[1] Department of Geothermal Science and Technology, Institute of Applied Geosciences, Technische Universität Darmstadt, Germany
[2] Darmstadt Graduate School of Excellence Energy Science and Engineering, Technische Universität Darmstadt, Germany
[3] APS Antriebs-, Prüf- und Steuertechnik GmbH, Göttingen Rösdorf, Germany

*Correspondence to*: Wolfram Rühaak (ruehaak@geo.tu-darmstadt.de, w.ruehaak@online.de)

**Abstract.** Performance and accuracy of machine learning techniques to segment rock grains, matrix and pore voxels, from a 3D volume of X-ray tomographic (XCT) grey-scale rock images was evaluated. The segmentation and classification capability of unsupervised (k-means, fuzzy c-means, self-organized maps),
supervised (artificial neural networks, least square support vector machines) and ensemble classifiers (bragging and boosting) was tested using XCT images of Andesite volcanic rock, Berea sandstone, Rotliegend sandstone and a synthetic sample. The averaged porosity obtained for Andesite (0.15 ± 0.017), Barea sandstone (0.15 ± 0.02), Rotliegend sandstone (0.14 ± 0.08), synthetic sample (0.50 ± 0.13) is in very good agreement to the respective laboratory measurement data and varies by a factor of 0.2. The k-means algorithm is the fastest of all
machine learning algorithms, whereas least square support vector machine is the most computationally expensive. Assessment of accuracy by entropy and purity values for unsupervised techniques; mean squared root error, receiver operational characteristics (to train the classification model) for supervised techniques; and 10-fold cross validation for the ensemble classifiers was performed. In general, the accuracy was found to be largely affected by the feature vector selection scheme. As it is always a trade-off between performance and
accuracy, it is difficult to isolate one particular machine learning algorithm which is best suited for the complex phase segmentation problem. Therefore, our investigation provides parameters that can help selecting the appropriate machine learning techniques for phase segmentation.

## 1 Introduction

Accurate segmentation of different phases from X-ray computer tomography (XCT) rock images is a well know
and complex problem in the digital rock physics community (DRP). The assessment and accurate segmentation of different phases from the XCT rock images is crucial for a reliable prognosis of the transport processes, elastic and electric properties; simulated using digital rock physics (DRP) models (Iassonov et al., 2009). The segmentation problem is reduced to the need to quantify the binary solid-void phase distribution (i.e., a binarization problem); choosing an appropriate scheme to binarize an image is key to characterizing a porous
space with a good degree of accuracy and further decreasing the magnitudes of the uncertainties involved in determining the geometries of pore networks (Leu et al., 2014). Therefore, machine learning (ML) techniques provide a promising alternative to segment and classify different voxel phases from XCT rock images. Based on





the heterogeneity of the sample the user can employ different ML techniques to obtain the best segmented image(s) which can be further used for simulating physical processes.

In Chauhan et al. (2016) a workflow was developed to segment XCT images using unsupervised, supervised and ensemble classifiers ML techniques (Figure 1). The focus of this study is to assess the performance and

accuracy of the above mentioned ML techniques to segment rock grain, matrix and pore phases in heterogeneous rock samples such as Andesite, Berea sandstone, Rotliegend sandstone and synthetic sample containing micro porosities.

## 2. Experimental Approach

For this study Andesite (Tongariro National Park, New Zealand), Berea sandstone (Andrä et al., 2012),

Rotliegend sandstone (Rotliegend Germany) and Synthetic sample Musli (provided by APS Antriebs, Prüf und Steuertechnik Gmbh Göttingen Rösdorf Germany) were used. Figure 2 shows the rock samples and respective histogram plots obtained from the XCT raw files. Effective porosity of Andesite ($17 \pm 2$ %) and Rotliegend sandstone ($14 \pm 2$ %) was measure using a GeoPyc pycnometer (Micromeritics Instrument Corporation Norcross, GA, USA). Thin section analysis using polarized microscope revealed Andesite has a porphyritic

texture with large plagioclase crystals (up to 3 mm in diameter), pyroxene in a cryptocrystalline matrix, and isolated vesicles up to 6 mm in diameter (Chauhan et al., 2016). Whereas, Rotliegend Sandstone had different grain size (between 0.5 to 5 mm) of fine sand and gravel, with monocrystalline quartz 26 %, poly-crystalline quartz up to 35 % , Feldspate 8 %, sedimentary volcanic lithoclast grains 9 % along with 13 % cement (Aretz et al., 2013). Andrä et al., 2012 confirms that the porosity of the Berea sandstone (total porosity 19.97 %; TM

Petroleum Cores Ohio USA) was performed using Helium Pycnometer 1330 (Micrometrics Instrument Corp. Belgium) and a mercury porosimetry using Pascal 140+440 Mercury Porosimeter (Thermo Electron Corporation, Germany). Madonna et al. (2012) scanning electron microscope revealed Berea Sandstone has Ankerite, Quartz, Zircon, K-spar and Clay. The Synthetic sample contained large pores, micro pores and mineral grain.

Andesite volcanic rock and Rotliegend sandstone where imaged using custom-built XCT scanner based on CT-Alpha system (ProCon, Sarstedt Germany) at the institute for Geoscience laboratory in Mainz Germany. The samples were scanned by applying X-ray energy of 110 keV and using a prefilter of 0.3 copper. During the reconstruction of the projections noise filter was not used. The projections were Radon-transformed in sinograms, thereafter converted through back-projection into tomograms. These stacked tomograms resulted in a

16-bit 3D imagery, with a resulting voxel resolution of 13 μm and 21 μm for Andesite and Sandstone respectively. Andesite required no beam hardening correction (BHC), whereas BHC for Sandstone was done based on regression analysis using 2D paraboloid fitting. Finally, the tomograms are saved in raw format.

The Berea sandstone dataset was obtained from GitHub FTP server (https://github.com/cageo/Krzikalla-2012). Andrä et al. (2012) performed XCT scans at tomographic microscopy and coherent radiology experiment

(TOMCAT) (Stampanoni et al., 2006) beamline at Swiss Light Source (Paul Scherrer Institute, Villigen, Switzerland). The beam energy was tuned for best contrast at 26 keV with an exposure time of 500 ms to retrieve a magnification of factor 10 (Andrä et al., 2012). The projections were magnified by microscope optics and digitized by high resolution CCD camera (PCO.2000), to obtain images of elements 1024 x 1024x 1024





with voxel resolution of 0.74 μm. Tomographic images were reconstructed from the by applying Fourier transform (Marone et al., 2009), were saved in desired file formats (Andrä et al., 2012).

**3 Machine learning and image processing**

The main focus of this study is to demonstrate the computational performance and accuracy of the different machine learning (ML) algorithm to segment/classify different phases in XCT rock samples - meaning, to map pixels of similar values in to respective classes. ML algorithms rely of features; features are a set of instances which contains descriptive information based on which the ML algorithm trains it classification model and further identifies these features in an unknown dataset and group them in to respective classes. Which in our case are the associated feature values of noise, rock grain, matrix and pore voxels. ML algorithms in general fall in to categories of unsupervised, supervised and ensemble classifiers.

**3.1 Unsupervised techniques**

In the unsupervised technique k-means (MacQueen, 1967), fuzzy c-means (FCM) (Dunn, 1973) and self-organized maps (SOM) (Kohonen, 1990) were used for segmentation pore, mineral and matrix phases. k-means is one of the simplest unsupervised ML algorithms commonly used to address clustering problem. The k-means algorithm through an iterative scheme calculates the Euclidean distance between the data point (pixel value) to its nearest centroid (cluster). The algorithm converges when the mean squared root error of Euclidean distance reaches minimum, that is, when no further pixel is left to be assigned to the nearest centroid (cluster). The performance of the k-means algorithm is strongly governed by the initial choice of the cluster centres. The k-means has the tendency to terminate without identifying the global minimum of the objective function (Chauhan et al., 2016). Therefore, it is recommended to run the algorithm several times to increase the likelihood that the global minimum of the objective function will be identified.

Unlike k-means, in the FCM iterative scheme each data point can be a member of multiple clusters by varying the membership function (Jain, 2010 and Jain et al., 1999). The FCM clustering procedure involves minimizing the objective function

$$J_{fcm}(Z;U;V) = \sum_{j=1}^{n} \sum_{i=1}^{k} (\mu_{ij})^m \left\| x_i^{(i)} - c_k \right\|^2 \tag{1}$$

where $c_k = \sum_{j=1}^{n} u_{ij} x_i$

Where $c_k$ is the $k^{th}$ fuzzy cluster centre, $m$ is the fuzziness parameter (for $m = 1$ FCM simplifies to k-means), $m. u_{ij}$ is the membership function. In our context, if we consider the entire raw image as a fuzzy set of data points (pixel values), which lie very close to each other - FCM uses membership criterial to "loosely" or "tightly" isolate subsets of rock grains, matrix and pore phase. Membership function influences the segregation of intersection subsets of values that lie in between rock grains/matrix phases for densely packed pixels (Rotliegend sandstone) and pore throat/matrix phases for micro pores dataset (Synthetic sample Musli). FCM can be a better choice in comparison to k-means; but it has a tendency to converge to the local minima of the objective function. Therefore, it is vital to test range of membership values in combination with several centroids (classes) for accurate analysis (Cannon et al., 1986).



For detailed description of SOM the reader is recommended to Kohonen (1990) and Chauhan et al (2016). SOM procedure uses a competitive learning process based on an artificial neural network framework (ANN). In our context, a raw CT image is considered as input pattern, which has to be classified. SOM first arranges nodes (called as neurons) in one of the desired topologies (grid, hexagon, or random topology; as specified by the user)

and assign random weight (values). These nodes are trained using the pixel value of the CT image(s), iteratively using Kohonen rule (Kohonen, 1990). During this competitive learning process the difference between the nodal weight and the neighbouring pixel(s) is calculated. The iterative process stops when the difference reaches to a minimum. The amount of adaptation of the nodal weight to its neighbouring values can be influenced and monitored using learning rate parameter $\alpha$. The nodes that do not change to it surrounding value are classed as

winner nodes. These winner nodes are nothing but different classes in the segmented image.

The unsupervised algorithms were configured to perform segmentation of three to seven classes. These classes in one-dimensional feature space are the non-overlapping segments of pixel bins in a histogram. Filter based feature vector selection (Euclidian and Manhattan distance function) were used to initialize centroids for k-means, FCM and SOM. In the case of FCM different degree of membership values [1.10 to 1.85] were tested to

'loosely' or 'tightly' segregate pixel values between rock grains and matrix phase. Grid topology was chosen in the case of SOM.

### 3.2 Supervised techniques

In the supervised category feed forward artificial neural Network (FFANN) (Jain et al., 1999) and least square support vector machine (LS-SVM) (Suykens and Vandewalle, 1999) were used to classify rock grains, matrix

and pore phases (Chauhan et al., 2016). In general, the supervised algorithms rely on a classification model which has to be trained using example set of data that represent each class.

ANN is an information processing paradigm that mimics the behaviour of the human brain (Haykin, 1994). FFANN is based on the ANN framework and uses so called error back propagation algorithm (Hopfield, 1982). FFANN can be used for any input-output mapping problem but is best suited for modelling linear and nonlinear

problems. In our case The XCT dataset was partitioned in to training and testing dataset. Thereafter, FFANN was setup with input layer, one hidden layer and output layer. The hidden layer was assigned 10 nodes, and the nodes of input and output layer varied depending on training and testing slices. The k-means, FCM segmented dataset were used as feature vector to train the classification model using Levenberg-Marquardt backpropagation method (Levenberg, 1944; Marquardt, 1963). The classification model was tuned using ten-

fold cross validation function (repeated trained and testing) and the misclassification rate was determined using mean squared root error (MSE). Once the classification model reached optimal accuracy it was tested on rest of XCT raw slices.

For LS-SVM a training data set was created, which contained range of pixel values which best represented pore, mineral, matrix and noise regions, these pixel ranges where further labelled in to different classes, which ranged

from one to seven. For FFANN and LS-SVM the models were tuned using ten-fold cross-validation function (repeated training and testing) and misclassification rate was determined using mean square root error (MSE) in the case of FFANN. Once the classification model reached an optimal performance threshold it was tested on rest of the XCT slices.



### 3.3 Ensemble classifiers techniques

In the ensemble classifier technique RUSBoost and Bragtree algorithms are used (Seiffert et al., 2008; Breiman, 1996) to classify pore, rock grains and matrix phases (Chauhan et al., 2016). In general ensemble classifiers are a 'bootstrap aggregation' of different weak classifiers. The main difference between Bragging and RUSBoost is

the way they train their weak classifiers. Bragtree is an iterative scheme, classifiers are trained with randomly chosen samples from the training data set, in the second step the misclassified instances are collected and its classifiers are retrained until the misclassification error is minimized. Whereas, RUSBoost sequentially trains its classifiers using the whole training set, essentially focusing on retraining inaccurate classifiers with the large data set until its misclassification error is minimized. The ensemble classifiers where trained using the same

feature vector (FV) which was used for LS-SVM, with a minimum leaf size of five and learning rate of 0.1.

### 3.4 Feature selection

In a practical rock CT segmentation/classification task a set of apriori information in the form of most useful pixel values is given to ML algorithms for segmentation or training the classification model. This dataset containing apriori information is termed as feature vectors (FV). For unsupervised k-means, FCM, SOM a set of

ten XCT images were used to develop the FV. For FFANN five images out of ten were used to train the network; for LS-SVM and ensemble based classifiers different subset of pixels representing the pore, mineral, matrix and noise regions were used as feature vectors. The total number of pixel used to train and test each ML algorithm is shown in Table. 1

### 3.5 Performance and Accuracy

Computational performance was measured in terms of the segmentation and classification speed of the ML algorithms. Test were performed on Windows Server 2008 R2 Standard 64-bit Operating System, with two six-core processor Intel Xenon, CPU (E645, 2.40 GHz) and installed memory (RAM) of 48.0 GB. For unsupervised techniques accuracy or cluster validation was studied to identify ideal class(es), representing the 'best' porosity values and to compare the clustering approaches. External validation measures 'Purity' and 'Entropy' was

performed on all the pixels corresponding to the classes' three to seven. The Purity and Entropy measure the ability of the clustering method to recover the know classes, despite number of classes are different from number of segmented classes (Jain et al., 1999). Purity is a real number between [0, 1], the larger the purity values, the better is the clustering method. Conversely, the lower the entropy value, the better is the clustering performance. In the case of FFANN, an objective method to determine the critical classification is by

calculating the mean square root error (MSE) between the output and the targets. The lower the MSE value, the better is the classification; zero corresponds to no misclassification. For LS-SVM receiver operation characteristics (ROC), a curve was plotted to compute the accuracy. This ROC curve gives the quality of the classification model. It shows a trade-off between the sensitivity of the classification model, with respect to the accuracy with which it can classify unknown data set. The area under the ROC curve represents the accuracy of

the classification model. The area of 1 represents a prefect classification; an area of 0.5 represents a misclassification (Khan et al., 2016). In case of Bragging and Boosting misclassification cost of the ensemble classifiers is estimated using 10 K fold cross-validation techniques.



## 4 Results and discussions

### 4.1 Porosity and pore size distribution

The porosities which were determined from the stack of ten XCT slices for three to seven classes using different ML techniques are shown in the Figure 3. The estimated porosity is the ratio between the pore phase voxels and entire sample volume multiplied by 100. In general, the porosity using unsupervised ML techniques agree well for all the four samples within ±1.2 % for each class. For Andesite, Berea, sandstone, Rotliegend sandstone and Musli, the averaged estimated porosity sum over all classes is 15.8 ± 2.5 %, 16.3 ± 2.6 %, 13.4 ± 7.4 % and 48.3 ± 13.3 % respectively. This is in good agreement to the experimental porosity values obtained for Andesite, Rotliegend sandstone using GeoPycpynometer and Berea Sandstone as reported in Ändra et al. (2012). The large standard deviation in the case of sandstone and Musli is caused by FCM segmentation scheme. When the membership function is tightly constrained [1.10, 1.35] the segregation between pore phase voxels and pore throat voxels is underestimated contributing to the increase in porosity. Conversely, when membership function loosely constrained [1.60, 1.85] pore throat and micro pores are segmented as matrix phases resulting in decrease in porosity and increase in matrix phase, which is clearly visible in the volume fraction plot of Sandstone and Musli in Figure 4. The low standard deviation in the estimated porosity values of Andesite is due to the absence of micro porosity and interconnected pores. The pore, mineral and matrix phases are distinct from each other therefore the ML techniques have less difficult in segmentation and classification. Figure 5 shows the segmented images using unsupervised technique and respective volume rendered images.

Pore size distribution (PSD) of Andesite, Sandstones and Musli was computed using the method suggested by Rabbani et al. (2014). The segmented grey scale images where first converted to binary images using thresholding technique. Morphological and filtering operations were performed based on the complexity of the segmented images. Distance transform to convert the bright area into catchment basin and later watershed transformation was performed to segment the pore boundaries. Figure 6 shows the PSD and average pore radius of Andesite, Berea sandstone, Rotliegend sandstone and Musli from k-means segmented images.

### 4.2 Performance and Accuracy analysis

Performance in the form of computational time is tabulated in Table 2. k-means algorithm is the fastest among all the ML techniques because segmentation of phases into different classes is based on nearest neighbourhood distances measurements; unlike other ML techniques (exception FCM), where the classification is governed by classification models.

In case of supervised techniques the computational speed is correlated to the size of the feature vector used for training the classification model and post processing of the unknown dataset. One reasons is that supervised techniques are based on a 'single' classification model; training and cross validation of the model with a large amount of feature vectors consumes time. This can be correlated to the high computational time of the Andesite sample using FFANN, were five slices were used to train the classification model compared to other samples where the classification model was trained using only one slice. For LS-SVM – as feature vector pixels are less than 1 % of the total pixel values for the all the samples – the training of the classification model took 1 to 10 minutes. The high computational time was consumed in post processing, large unknown dataset using the trained model. In the case of ensemble classifiers the post processing of an unknown dataset took longer compared to the training of the respective (bootstrapped weak) classification schemes. As the Rotliegend





sandstone is densely packed with very low porosity, it resulted in low contrast and badly resolved XCT dataset. As a consequence, the individual (weak) classification models required more computational time to reach to a consolidated nearly accurate well classified result. Therefore, the processing time of Rotliegend sandstone images by ensemble classifiers was higher compares to other XCT samples.

Our clustering problem is to determine the most appropriate class for each phase. That is, we wish to identify which of the unsupervised ML technique satisfies properties of "cluster homogeneity" (i.e. not mixing items belonging to different categories) and "cluster completeness" (i.e. how good items belonging to same categories are group together) defined by Amigó et al. (2008). Therefore, the metrics entropy and purity were chosen to evaluate the accuracy of unsupervised ML techniques. The entropy values were calculated using 3D stack of ten

slices for each class and are shown in Figure 7. In general class three and four have the lowest entropy values compared to other classes. This shows that if cluster homogeneity is over-segmented and cluster completeness gets violated this may lead to misclassification. Among the three unsupervised ML techniques, k-means has the lowest entropy values therefore it can be assumed that k-means performs the best segmentation compared to SOM and FCM.

For FFANN the accuracy was interpreted using the MSE error shown in Figure 8. FFANN was trained using k-means and FCM and was tested on raw XCT images of the respective samples. The testing dataset (3D stack of raw images) was scaled between three to seven class values before the start of the testing cycle. In the case of Berea, Rotliegend and Synthetic sample, when the membership function was tightly constrained to 1.10, FCM was able to segment, pore, matrix and mineral grain phases into maximum of three and four classes. Similarly,

on moderate (1.60) and loose constrained (1.85) membership function FCM yield maximum of five, six and seven classes respectively. This explains the variance in the number of dataset used for validation of FFANN. The lower the MSE value, the better is the accuracy; the accuracy decreases with over classification (for class five to six). Different settings such as, increase of the number of training slices up to five and increasing the number of neuron from ten to thirty did not shown any significant improvement in the accuracy. Among all the

XCT samples, the worst accuracy was found for Rotliegend sandstone. Based on our analysis, we suggest that FFANN may not be the best suited ML technique for clustering analysis.

In the case of LS-SVM, the low variance seen in the porosity values up to class six, is the indication that LS-SVM is one among the most suitable ML technique for phase segmentation analysis of XCT images. As the hand-picked feature vector dataset of class four had an appropriate mix of all the phases and desired amount of noise, it gave the best trade-off between quality and speed. Hence we show the accuracy of LS-SVM for

classification of class four using ROC curve (Metz, 1978) in Figure 9. The slope of the ROC curve gives the accuracy of classification. The accuracy ranges between 77 % for Berea sandstone, 88 % for Rotliegend sandstone and 90 % for Andesite and Synthetic sample Musli. Up to 100 % accuracy in achieved in discriminating the pore phase with respect to mineral and matrix phases.

Ensemble classifiers also show low variance in the porosity values as LS-SVM because of the same feature

vectors used. The accuracy of the ensemble classifiers were tested using 10 K fold validation technique (Quinlan, 1996) is shown in Figure 10. Both Bragging and Boosting classifiers where trained using the training data set. The training dataset comprises of the pixel values representing pore, mineral, matrix, noise phases and feature vectors. The initial growth of the leaf size was started with five and the corresponding weak classifiers

were trained up to thousand iterations. On the onset of the 10 K fold validation procedure the training dataset





was mixed in a random order, thereafter the dataset was partitioned in to chucks of 10 blocks. Using a for-loop with an increment of from one to ten, the classifier was trained with the examples which did not belong to the $i^{th}$ fold and tested with the examples of the $i^{th}$ fold. The accuracy was determined by computing the mean square root error of number of pixels which were wrongly classified to the total number of pixels. The best accuracy

was achieved for Andesite and Musli XCT (with an exception for class six) images and the worst for Rotliegend sandstone going up to 0.56.

**Conclusions**

In this study the performance and accuracies of ML techniques were validated and relative porosity and pore size distribution of Andesite (altered minerals), Berea sandstone, Rotliegend sandstone (inter connected pores)

and Musli (micro porosity) rock samples were computed. The total averaged porosity values obtained using unsupervised, supervised and ensemble classifiers are shown in Figure 11 and are in good agreement with the experimental values obtained using GeoPycpynometer and data reported in Ändra et al. (2012). The high standard deviations up to 13 % seen in the case of Synthetic sample Musli can be attributed to the misclassification caused by ensemble classifiers in class six, seen in Figure 3.

Our analysis shows unsupervised ML techniques perform well with filter based feature extraction techniques. In terms of computational time, k-means outperforms all the other ML techniques. Fuzzy c-means can distinguish well between pore and pore-throat boundaries, given that the membership function is loosely constrained between 1.60 - 1.85. It was found that different tuning parameters (such as different FCM membership criteria and different SOM topologies and distance functions) need to be tested for the unsupervised techniques. A SOM

topology "gridtop" layout (neurons arranged in a grid format) and a SOM Manhattan distant function (sum of the absolute difference) gave consistent results and FCM membership function between [1.35 - 1.85] gave consistent results. Low entropy values of k-means indicates that k-means is more accurate compared to fuzzy c-means and self-organized maps.

In the case of supervised techniques the computational time was significantly improved by reducing the training

dataset of feed forward artificial neural networks (FFANN) and by careful selection of feature vector dataset for least square support vector machine (LS-SVM). Based on our analysis we conclude that FFANN may not be best suited for clustering analysis; due to difficulty in scaling the training dataset (XCT raw files), the interpretation of clustering labels and accuracy becomes extremely difficult. Additionally, the accuracy in terms of mean square root error of the validation cycle (training and repeated testing) is largely regularized by fine and

coarse scaling of the testing dataset, which may not always correspond to the image classification. As a consequence, there were cases where despite low accuracy (high MSE error) the classification performed by FFANN was good. On the contrary LS-SVM showed to be one of the best and accurate supervised ML technique for phase segmentation problem. However, it strongly relies on the craft with which the feature vector dataset is constructed. The user has the flexibly to decide which phases or feature are most relevant for phase

segmentation. The authors suggest using the histogram plot of the raw image or k-means (or any other unsupervised ML technique) as an orientation for feature vector selection. It is further recommended that the first and second class labels (ex. class three and class four) should contain predominantly phases such as pore, matrix, mineral and noise pixels. Consequently, other interesting feature pixels can be included. A suitable balance has to be found, such that the classifier is not excessively trained on one particular feature and get stuck





in local minima. Thereafter, the receiver operation characteristic (ROC) curve validation technique is best suited for accuracy assessment of LS-SVM.

Ensemble classifier can be the second best alternative to tackle phase segmentation problems as it also relies on the feature vector dataset to train the classification model; therefore, the user has more control over the classification scheme. However, the weak learners involved in the ensemble classification scheme remain as black-box to a large extent; therefore, appropriate tuning of the individual weak learners to optimise computational speed and accuracy may be cumbersome. To have a better control over the ensemble classification scheme, and for future work we suggest an ensemble classifier with k-means, FCM and LS-SVM as weak learners.

*Acknowledgments*

We thank Michael Kersten, Frieder Enzmann and his group at the institute for Geoscience Johannes Gutenberg Universität Mainz for the high resolution X-ray tomography measurements of Andesite and Rotliegend sandstone. We thank Phillip Mielke and Achim Aretz for the field work and collection of rock samples. Hieko Andrä and her team at Fraunhofer ITWM Germany for making synchrotron dataset of Berea sandstone freely available online. This work is also partly supported by the DFG in the framework of the Excellence Initiative, Darmstadt Graduate School of Excellence Energy Science and Engineering (GSC 1070) and APS Antriebs- , Prüf- and Steuertechnik GmbH, Rosdorf, Germany.

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





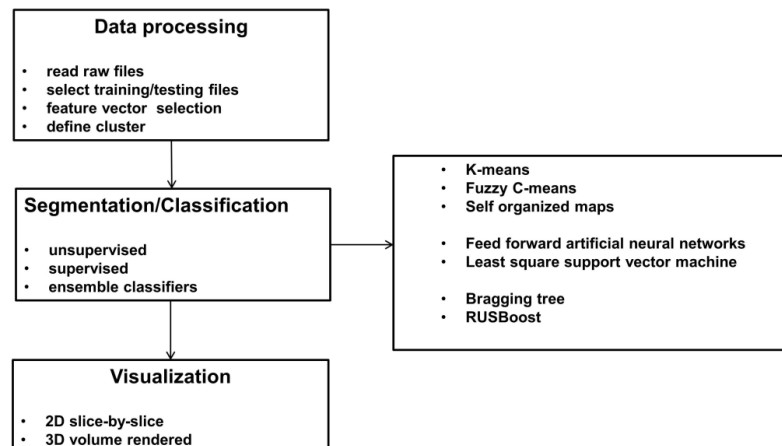

**Figure 1.** Schematic diagram of the work flow (after Chauhan et al., 2016).

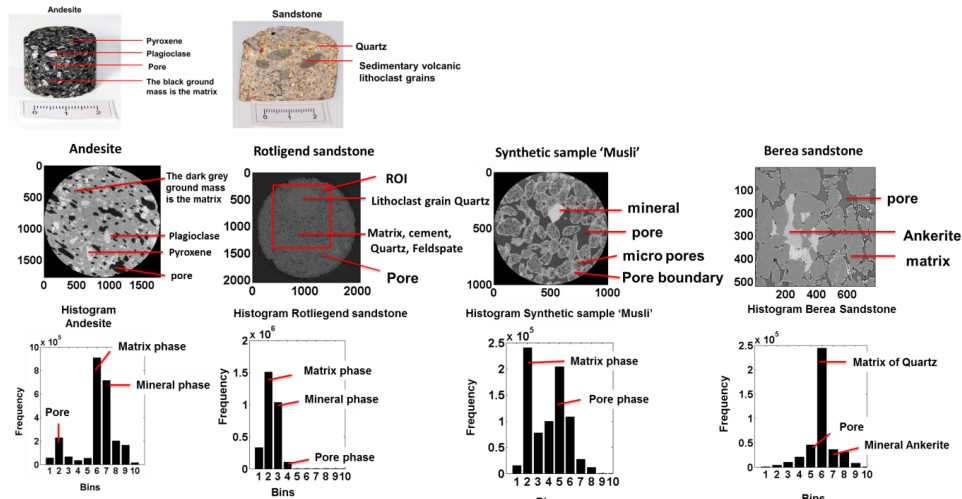

**Figure 2.** The top panel shows the Andesite and Rotliegend sandstone rocks used for XCT measurements. Middle panel
5 shows the raw images of Andesite (16bit), Rotliegend sandstone (16 bit), synthetic sample (16 bit) and Berea sandstone (16 bit). Mineral composition of Andesite and Rotliegend sandstone was determined from thin sections using polarized microscope. Bottom panel shows, histogram plot of the respective samples. Mineral composition of Berea sandstone is based on Madonna et al. (2012) and Andrä et al. (2013).



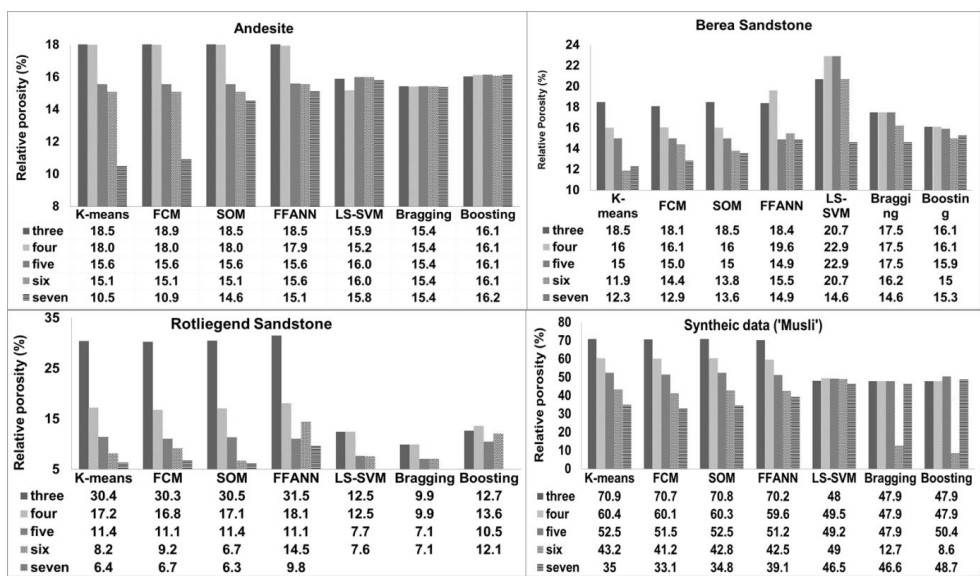

**Figure 3.** Relative porosity values obtained using unsupervised, supervised and ensemble classifier techniques for respective samples.

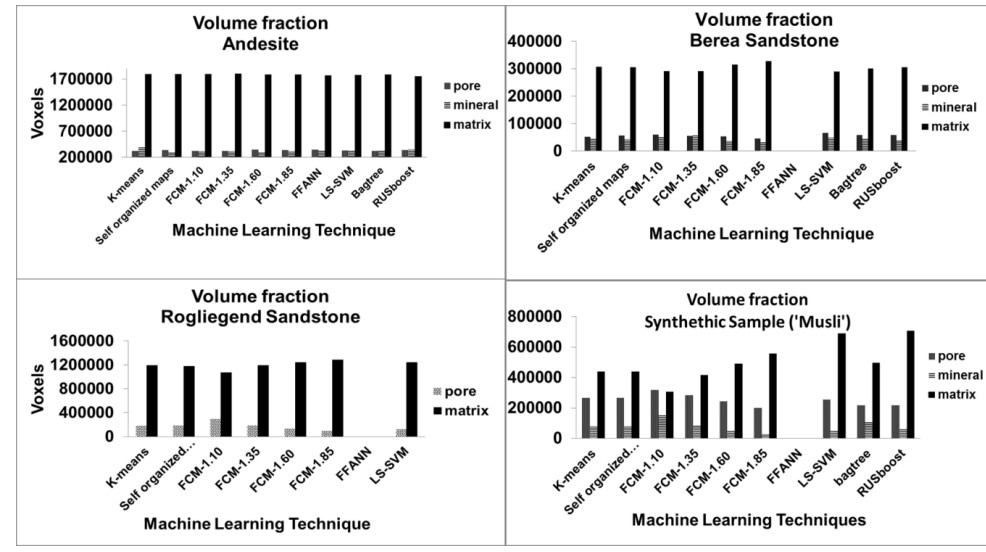

**Figure 4.** Total volume fraction plotted for respective samples









**Figure 5.** The top, middle and last panel show the 2D segmented images and volume rendered plots of respective samples using unsupervised networks (Andesite figure has been modified after Chauhan et al 2016).





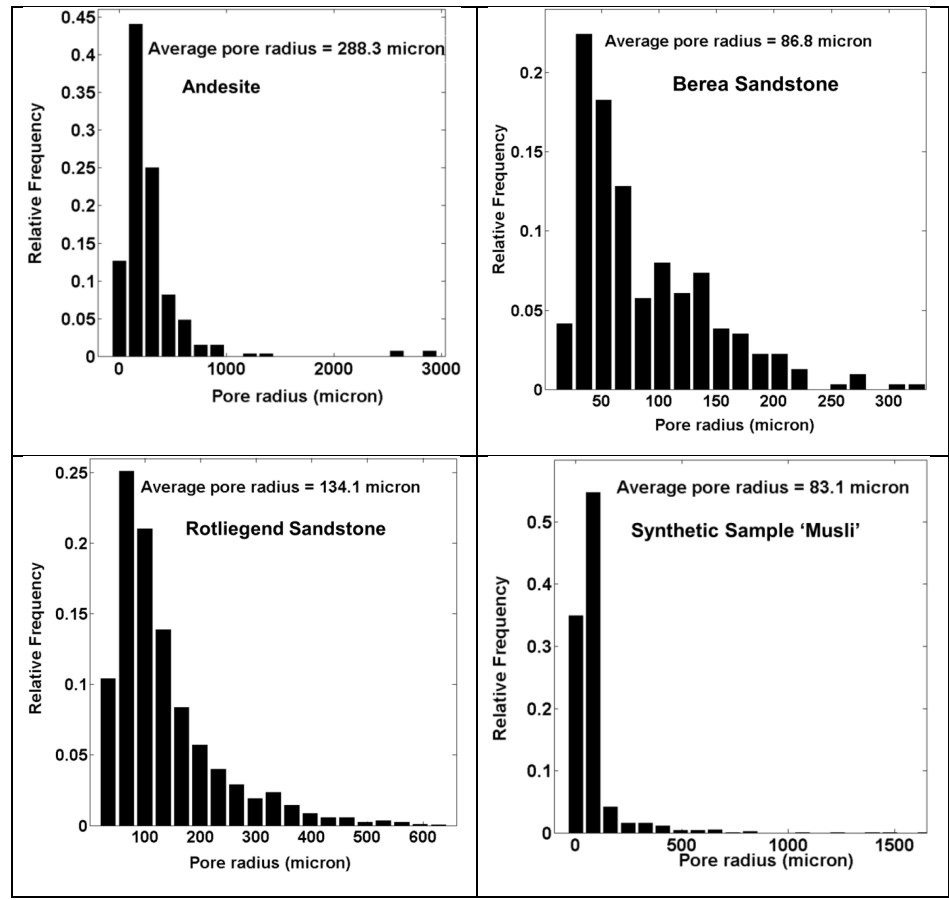

**Figure 6.** The right panel shows relative porosity using machine learning algorithms for different rock samples. Middle panel shows the volume fraction of different phases quantified using machine learning techniques and the right panel show
5    the pore size distribution of different sample using watershed technique.




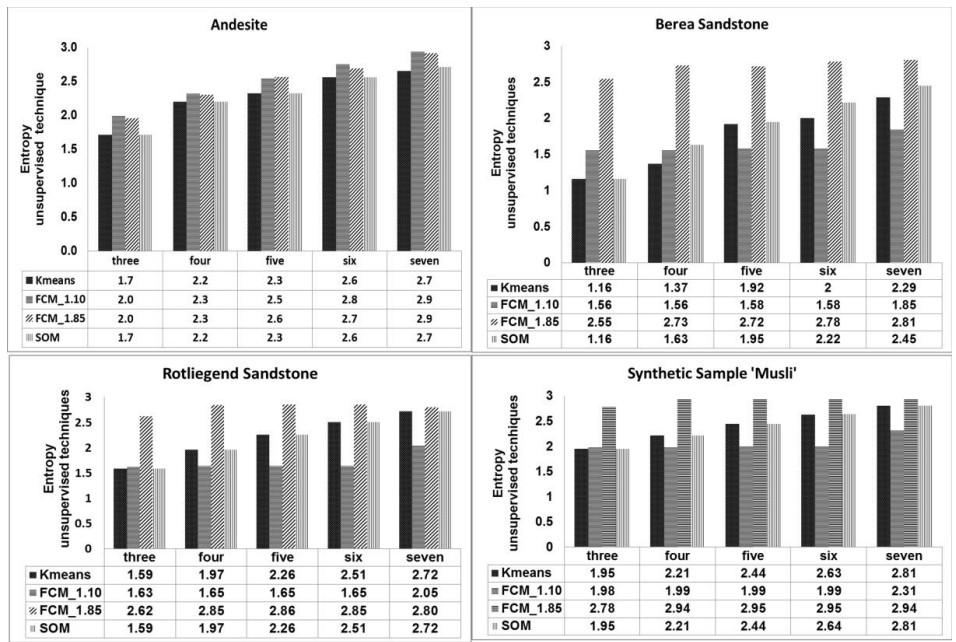

**Figure 7.** Entropy values of unsupervised techniques plotted for respective samples.

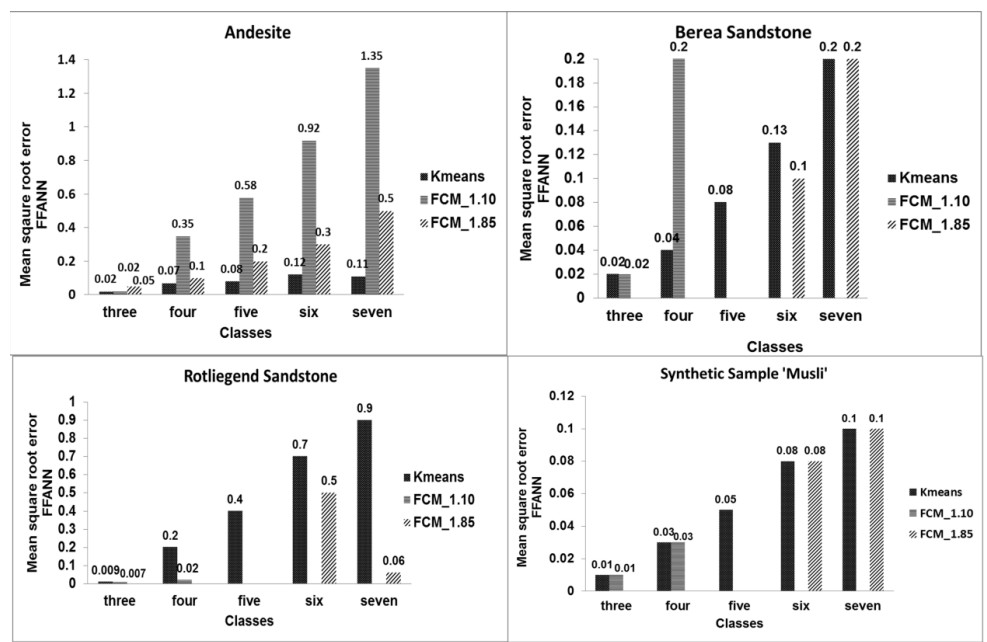

**Figure 8.** Mean square root error values of feed forward artificial neural network obtained for respective samples.




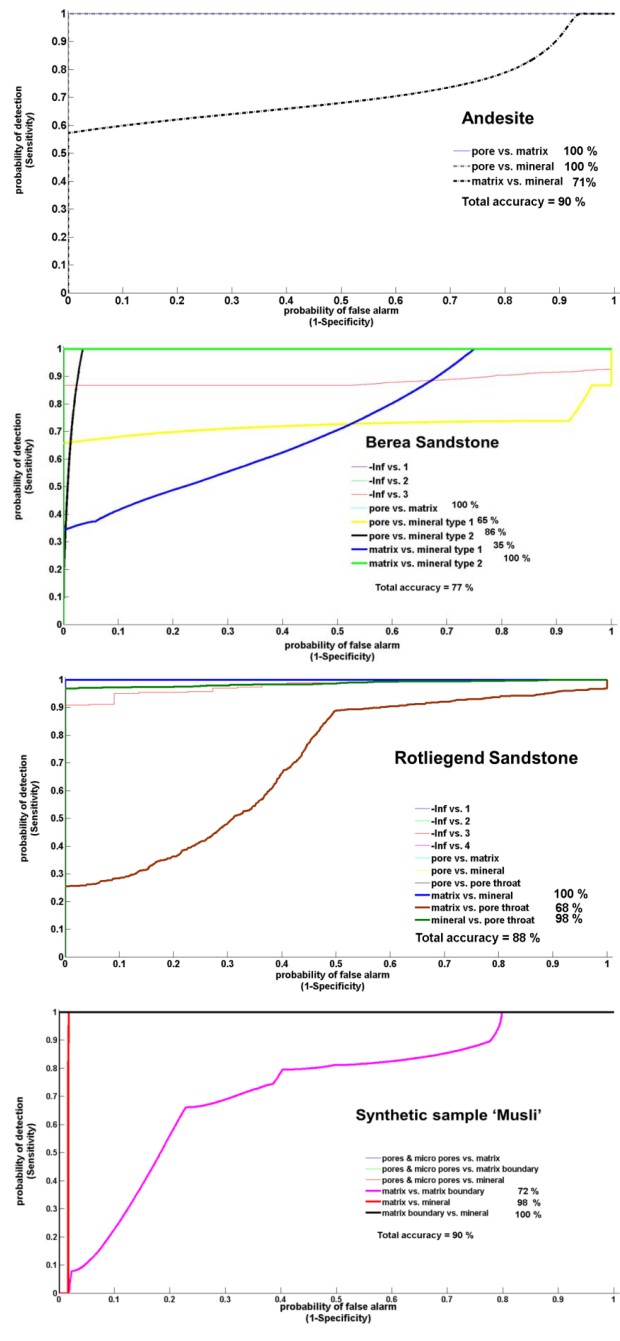

**Figure 9.** Receiver operational characteristic curves depicting the accuracy of least square support vector machine multi classification scheme for class four.





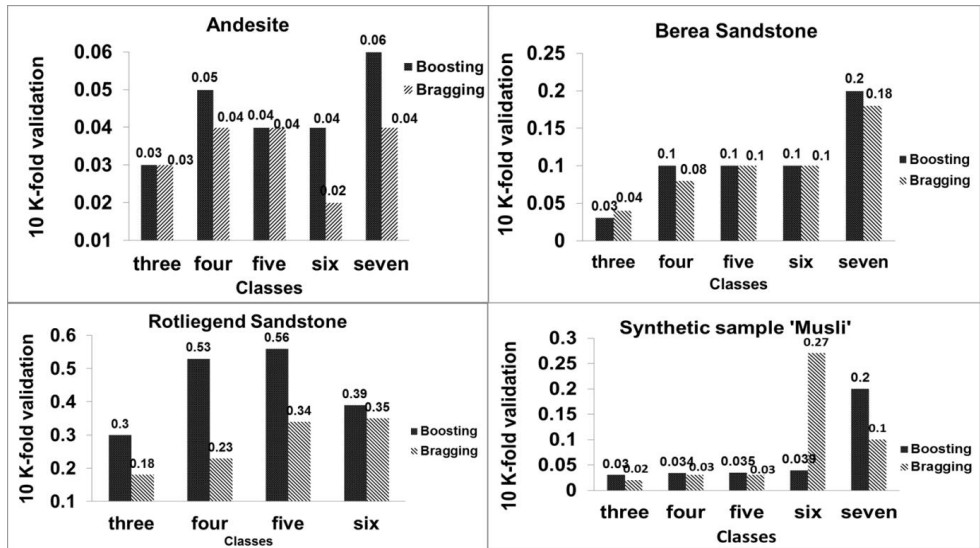

**Figure 10.** Accuracy of ensemble classifiers Boosting and Bragging calculated using 10 K-fold validation for respective samples.





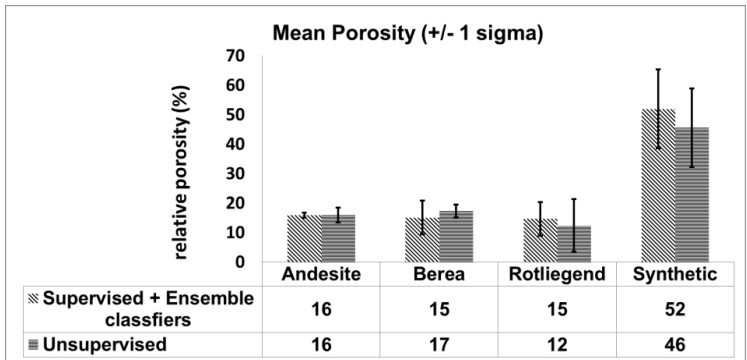

**Figure 11.** Mean porosity value obtained using supervised, ensemble classifiers and unsupervised machine learning techniques.

5    **Table 1. T**he number of pixels used for training and testing the classification model.

| Type of classifiers | Andesite | | Rotliegend Sandstone | |
|---|---|---|---|---|
| | No. of training pixels | No. of testing pixels | No. of training pixels | No. of testing pixels |
| k-means | | 31,577,290 | | 13,681,600 |
| fuzzy c-means | | 31,577,290 | | 13,681,600 |
| self-organized maps | | 31,577,290 | | 13,681,600 |
| artificial neural networks | 15,788,645 | 31,577,290 | 6,840,800 | 13,681,600 |
| least square support vector machine | 2077 | 31,577,290 | 1511 | 41,943,040 |
| Bragging and Boosting | 2077 | 31,577,290 | 1511 | 41,943,040 |
| | Synthetics sample (Musli) | | Berea sandstone | |
| | No. of training pixels | No. of testing pixels | No. of training pixels | No. of testing pixels |
| k-means | | 10,000,000 | | 40,56,000 |
| fuzzy c-means | | 10,000,000 | | 40,56,000 |
| self-organized maps | | 10,000,000 | | 40,56,000 |
| artificial neural networks | 5,000,000 | 10,000,000 | 20,28,000 | 40,56,000 |
| least square support vector machine | 1655 | 10,000,000 | 1366 | 40,56,000 |
| bragging and boosting | 1655 | 10,000,000 | 1366 | 40,56,000 |





**Table 2.** Show the computational time for processing ten slices.

| Machine learning Techniques | CPU: Time (hrs:min:sec) | | | |
|---|---|---|---|---|
| | Andesite | Rotliegend sandstone | Synthetic sample Musli | Berea sandstone |
| K-means | 00:15:35 | 00:12:04 | 00:10:59 | 00:05:33 |
| FCM | 00:29:19 | 00:56:03 | 00:42:21 | 00:41:05 |
| SOM | 01:07:06 | 1:41:47 | 01:11:23 | 00:33:32 |
| FFANN (training using K-means) | 08:58:18 | 00:11:50 | 00:10:40 | 00:11:12 |
| LS-SVM[a] | 63:29:35 | 03:22:58 | 03:02:15 | 01:45:17 |
| Bragging | 05:57:05 | 07:32:22 | 12:19:40 | 03:51:13 |
| Boosting | 07:47:05 | 09:52:56 | 06:14:58 | 03:20:42: |

a open source public library provided by K.U. Leuven university –ESAT department- SCD-SISTA division was used. http://www.esat.kuleuven.be/sista/lssvmlab/