# Peer review of "Phase Segmentation of X-Ray Computer Tomography Rock Images using Machine Learning Techniques: an Accuracy and Performance Study"

_Solid Earth, 2016_

## Referee Comment (RC1) · Anonymous Referee #1 · 10 Apr 2016

Image segmentation is the most crucial step in image processing of micro-CT images of porous rocks, because functional properties are usually not derived from grayscale data itself but from the segmented data. For instance, Lattice-Boltzmann simulations are conducted on the segmented pore space, statistical analysis are carried out for different material classes, etc. A multitude of different segmentation exists to date, which differ vastly in computational complexity, underlying rationale and so on. There are many review papers on image segmentation, that try to sort these existing segmentation algorithms according to their methodological approaches or rank them according to their suitability for a set of test images. The general conclusion is often, that no segmentation excels all others in all cases and it depends very much on the image content

which one is suited best. Chauhan et al. wrote yet another of these review articles and limit their focus on machine learning algorithms, which might be well established in remote sensing, life sciencces or other scientific disciplines, but have not yet gained much attention when it comes to micro-CT images of rocks. Therefore, the article could in principle be useful in closing that gap. However I cannot recommend its publication for several reasons.

First of all, the article by Chauhan et al. can be understood as a follow-up study to Chauhan et al. (2016): Computers & Geosci., doi:10.1016/j.cageo.2015.10.013. The general purpose of the current study is again to test the suitability of various machine learning algorithms (supervised or unsupervised) to segment micro-CT images on a set of different rock images. The overlap with the precursor study is high. In fact, the only salient difference between them is that four images have been used for testing instead of one and I'm wondering why this hadn't been done in this first place.

There are several other issues with this paper:

1. The choice of different validation methods for different methods hampers comparability among all methods. For one method you use MSE, for another purity and entropy and for yet another method ROC curves are computed. Moreover, these validation methods need to be explained in much more detail including formulas. This includes MSE, ROC curve, 10K cross validation, purity and entropy metric.

2. Wording is frequently mixed with unexplained jargon and often does not meet a sufficient standard to follow the line of argument. I list plenty of examples below.

3. Introduction is too short. The introduction is only half a page long and doesn't barely touch the current state of knowledge.

4. Preprocessing and/or postprocessing is not discussed. Therefore, the suggested work flow is far away from common practice of most scientists, working in the field. For instance, the sandstone image seems to be extremely noisy. Therefore, most

colleagues would probably use some noise filter as a preprocessing step, or use some spatial regularization during image segmentation, e.g. by applying a locally-adaptive method, or apply some post-processing, e.g. majority filter or morphological operators, to clean up the results.

5. Conclusions are weak. There is no real line of argument in the conclusions. The findings are mainly reported for each method independently and are sometimes too specific (explanation below) to be generalized into something useful. In turn, some conclusions are basically what seems to be common sense, e.g. feature vectors have to be chosen carefully or there should only be as many classes as there are real phases in the image (not more, not less).

5. The machine learning algorithms seem to be very impractical for realistic datasets due to excessive computation time. The dataset were actually quite small, up to 31 megapixels (MP). Real micro-CT datasets nowadays have dimensions up to 8000MP. I could not find a comment on how CPU:time scales with image size, but it would definitely render LS-SVM as one of the recommended methods useless for most practical applications. This would leave k-means and FCM (p9l7-9) as the only recommended ML methods, and these exists already for more then four decades. All in all, I'm not convinced why I should use machine learning algorithms for micro-CT image segmentation in the future.

Technical comments:

p1l22-23: Bad wording

p2l38: Bad wording: ... to obtain images of elements 1024 x 1024 x 1024 ...

p3l1: Bad wording: ... from the by applying Fourier ...

p3l6: Bad wording: ... rely of features ...

p5l10: Meaning unclear: "... minimum leaf size of five and learning rate of 0.1."

p5l12: Meaning unclear: "... apriori information in the form of most useful pixel values." How can a pixel value be useful? Do you mean, that a pixel value is most likely to be assigned to a specific material, because according to the manually chosen feature vectors it is more similar to the class statistics of this material than any other material?

p5l14-15: meaning unclear: "... a set of ten XCT images". Do you mean ten slices from a XCT image?

p5l26-27: meaning unclear: "... the know classes, despite number of classes are different from number of segmented classes."

p5l30: "... between output and targets." Be more specific. What are targets? Class assignments for manually selected pixels? Are feature vectors the same as targets?

p5l33-34: meaning unclear: "It shows a trade-off between sensitivity ..." This sentence is a stub. Trade-off between sensitivity and what?

p5l35: typo: prefect

p6l6-9: What about unresolved porosity below the image resolution? Shouldn't the image-derived porosities be all much lower than the experimental porosity values, because they do not capture very small pores?

p6l10-15: Okay, so a higher fuzziness parameter shifts the pore/matrix threshold towards lower gray values, so that the volume fraction of pores is reduced? But why exactly is this the case? Also, in the conclusions (p8l16-18) you state that somehow FCM can distinguish between pores and pore-throats, which is not true, because it would mean that the algorithm could distinguish between different pore sizes or functional units of the pore space. All FCM does, is to evaluate the histogram (only grayscale information, no spatial information) and depending on how you set the fuzziness parameter, partial volume voxels are assigned to pores or matrix. Similar statement in p8l16-17.

p6l21-22: meaning unclear: "Morphological and filtering operations were performed

based on the complexity of the segmented images". Which image was processed how?

p6l31: meaning unclear: "... and post processing of the unknown dataset." This is the first time you mention post processing. Do you mean with "post", that it is carried out after some tentative image segmentation is completed? What do you do specifically during post-processing? Why is it somehow linked to the size of feature vectors?

p7l2: meaning unclear: "As a consequence, the individual (weak) classification models". What do you mean by weak?

p7l5: meaning unclear: "... most appropriate class for each phase." Do you mean: ... for each pixel?

p7l11: meaning unclear: " ... cluster homogeneity is over-segmented ..." This makes no sense to me.

p7l28-30: "As the hand-picked ... quality and speed". This sentence can probably only be understood by an absolute insider. Which material in which rock did you hand-pick to represent class four? Why do you consider a mix of all phases and noise appropriate? For me this actually sounds like the method failed completely, when class four does not represent a single material.

p7l39: meaning unclear: " The initial growth of the leaf size ...".

p8l1: typo: chucks

p8l1-3: meaning unclear: "Using a for-loop with an increment of from one to ten, ... ith fold." Since you did not describe 10K cross validation in the first place, it is not possible to understand this sentence without background knowledge.

p8l8-14: This paragraph should be part of the discussion and not the conclusions.

p8l14: What is class six? Why is class six different from others?

p8l19-22: This conclusion is highly specific to the SOM method and cannot be generalized into something useful. You're basically saying the the parameters that you've chosen a priori worked well. But maybe a hexagonal topology and a Euclidean distance function would have also done the job?

p8l26-30: Here in the conclusions you refer for the first time to "scaling". Why would you want to scale your images, when you're only interested in the segmentation results at the scale at which you acquired the image. Often a segmentation result at a coarse scale is of very little use. Therefore, the sentence "Additionally, the accuracy ..." is hard to follow.

p8l35-39: Of course the class labels should contain only one real phase in your rock, not only parts of it and not many simultaneously. Why is that an important conclusion?

Fig. 1: directly copied from older paper. It's not clear from the text, why this figure needs to be added.

Fig. 2: Why only show ten grayscale bins, when the original data is at least 8-bit, i.e. 256 bins?

Fig. 5: The legend suggest that there are also classes/colors with half labels, e.g. 0.5, 1.5, 2.5? Is this really the case?

Fig. 6: Figure caption is out of context.

Fig. 9: Fonts are too small. ROC curves are not self-explanatory. What does "-inf vs. 1" for Berea sandstone mean? What is probability of false alarm and probability of detection? How do you derive total accuracy from the individual curves?
* * *

---

## Referee Comment (RC2) · Anonymous Referee #2 · 27 Apr 2016

Dear authors,

from my point of view your contribution is not significant enough to be published in Solid Earth. The main reason is the only marginal added value to your already published paper "Processing of rock core microtomography images: Using seven different machine learning algorithms" (Computer & Geosciences, 2016). Your statement in the abstract "Therefore, our investigation provides parameters that can help selecting the appropriate machine learning techniques for phase segmentation" is rather vague to support the idea to have another paper to your already published paper.

Moreover, there are other very weak points in the paper:

[Figure]

- The introduction is too short and will give not a good overview of the topic. Do the authors will really not have a better overview?

- Other segmentation techniques are not discussed in the manuscript. Maybe they are superior to the discussed seven methods.

- Nowadays it is a standard to apply several filters in the data processing workflow. They are not mentioned or discussed.

- The quality of the figures is bad. Partly they are too small too identify details (e.g. Figure 2, 3 or 9) or the labeling is not explained (what is FCM-1_35 in Figure 4 ???)

In conclusion, the paper is not acceptable in its current form. There are simply too many things to correct.

---

## Author Comment (AC1) · 27 May 2016

Dear Editor:

We appreciate the efforts you and the reviewers have invested in our manuscript. We have addressed all relevant concerns of the reviewer and are submitting a revised version for your consideration.

We are aware, that the new content, with respect to the in both reviews mentioned paper of Chauhan et al., 2016, is only limited.

Notwithstanding of this, the paper we have submitted to SE is a completely new contribution with no redundancy to the previous one.

Also, we believe that it would be a nice contribution for this SI to round up things.

With respect to the technical comments we have revised the paper in such a way that the reviewer's comments are taken into account.

Following is an itemized list of reviewers' comments together with our response. Comments are reported in italic fonts (red) and our responses in regular (blue).

Sincerely,

On behalf of all authors, Wolfram Rühaak

*Anonymous Referee #1*

*Review for: Swarup Chauhan et al. "Phase Segmentation of X-Ray Computer Tomography Rock Images using Machine Learning Techniques: an Accuracy and Performance Study"*

*Overview:*

*Image segmentation is the most crucial step in image processing of micro-CT images of porous rocks, because functional properties are usually not derived from grayscale data itself but from the segmented data. For instance, Lattice-Boltzmann simulations are conducted on the segmented pore space, statistical analysis are carried out for different material classes, etc. A multitude of different segmentation exists to date, which differs vastly in computational complexity, underlying rationale and so on. There are many review papers on image segmentation, that try to sort these existing segmentation algorithms according to their methodological approaches or rank them according to their suitability for a set of test images. The general conclusion is often, that no segmentation excels all others in all cases and it depends very much on the image content which one is suited best. Chauhan et al. wrote yet another of these review articles and limit their focus on machine learning algorithms, which might be well established in remote sensing, life sciencces or other scientific disciplines, but have not yet gained much attention when it comes to micro-CT images of rocks. Therefore, the article could in principle be useful in closing that gap. However I cannot recommend its publication for several reasons.*

*First of all, the article by Chauhan et al. can be understood as a follow-up study to Chauhan et al. (2016): Computers & Geosci., doi:10.1016/j.cageo.2015.10.013. The general purpose of the current study is again to test the suitability of various machine learning algorithms (supervised or unsupervised) to segment micro-CT images on a set of different rock images. The overlap with the precursor study is high. In fact, the only salient difference between them is that four images have been used for testing instead of one and I'm wondering why this hadn't been done in this first place.*

In fact there is no overlap. In the paper submitted to Solid Earth a completely new study is presented. Here the focus is on qualitative assessment (validation/accuracy studies) of three different machine learning techniques (ML).

Such a study required a separate dedicated case study and did not fit into the scope of the paper Chauhan et al. (*doi:10.1016/j.cageo.2015.10.013).*

*General Comments:*

*The choice of different validation methods for different methods hampers comparability among all methods. For one method you use MSE, for another purity and entropy and for yet another method ROC curves are computed. Moreover, these validation methods need to be explained in much more detail including formulas. This includes MSE, ROC curve, 10K cross validation, purity and entropy metric.*

Reply: the above mentioned metric are the commonly used validation methods for supervised, unsupervised and ensemble classifiers. We have unified the presentation of result evaluation and have added the respective equations.

Revised (P7-P9|5-14

*Wording is frequently mixed with unexplained jargon and often does not meet a sufficient standard to follow the line of argument. I list plenty of examples below.*

*Introduction is too short. The introduction is only half a page long and doesn't barely touch the current state of knowledge.*

Revised: P1-P3|29-4

*Preprocessing and/or postprocessing is not discussed. Therefore, the suggested work flow is far away from common practice of most scientists, working in the field.*

Revised: P4|6-12

*For instance, the sandstone image seems to be extremely noisy. Therefore, most colleagues would probably use some noise filter as a preprocessing step, or use some spatial regularization during image segmentation, e.g. by applying a locally-adaptive method, or apply some post-processing, e.g. majority filter or morphological operators, to clean up the results.*

Reply: In the case of Rotliegend Sandstone (21 µm) as the XCT images were noisy, contrast filter was used to enhance the image. Whereas, for other XCT images (Berea, Andesite and Musli) as the resolution and contrast were sufficiently high (7.5 µm to 13 µm) using filters did not show any noticeable change.

Revised: P4|6-12.

*Conclusions are weak. There is no real line of argument in the conclusions. The findings are mainly reported for each method independently and are sometimes too specific (explanation*

*below) to be generalized into something useful. In turn, some conclusions are basically what seems to be common sense, e.g. feature vectors have to be chosen carefully or there should only be as many classes as there are real phases in the image (not more, not less).*

Reply: We disagree. Within the scope of the conclusion, we have pointed out which of the ML algorithms are suitable for segmentation and classification and their respective advantages and disadvantages based on their accuracy and performance. In validation case studies of ML techniques it a common to suggest the best practice to perform accurate analysis.

*The machine learning algorithms seem to be very impractical for realistic datasets due to excessive computation time. The dataset were actually quite small, up to 31 megapixels (MP). Real micro-CT datasets nowadays have dimensions up to 8000MP. I could not find a comment on how CPU:time scales with image size, but it would definitely render LS-SVM as one of the recommended methods useless for most practical applications. This would leave k-means and FCM (p9l7-9) as the only recommended ML methods, and these exists already for more then four decades. All in all, I'm not convinced why I should use machine learning algorithms for micro-CT image segmentation in the future.*

Reply: Yes, some of the ML techniques require excessive computational time, mainly because we process gray scale values of large dimension (1024X1024x10 slices), instead of certain ROI. Whereas, commercial software such as GeoDict, AVISO use ROI combined with binary segmentation algorithm to optimize speed. So the comparison made by the reviewer is not valid.

Furthermore, if XCT and SEM images are combined using suitable image registration techniques, the multi-classification provided by supervised schemes can assist in clustering and mineral identification. Finally given the computational power available these days, considering ML for XCT segmentation is a good investment.

It is not clear what the reviewer means by— on how CPU:time scales with the image size ?

Not revised.

*Technical comments:*

*p1l22-23: Bad wording*

Revised: P1|22-24

*p2l38: Bad wording: ... to obtain images of elements 1024 x 1024 x 1024 ...*

Revised: P4|2

*p3l1: Bad wording: ... from the by applying Fourier ...*

Revised: P4|3-4

*p3l6: Bad wording: ... rely of features ...*

Revised: P4|16

*p5l10: Meaning unclear: "... minimum leaf size of five and learning rate of 0.1."*

Revised: P6|22-28

*p5l12: Meaning unclear: "... apriori information in the form of most useful pixel values." How can a pixel value be useful? Do you mean, that a pixel value is most likely to be assigned to a specific material, because according to the manually chosen feature vectors it is more similar to the class statistics of this material than any other material ?*

Revised: P6|30-34

*p5l14-15: meaning unclear: "... a set of ten XCT images". Do you mean ten slices from a XCT image?*

Revised: P5|23

*p5l26-27: meaning unclear: "... the know classes, despite number of classes are different from number of segmented classes."*

Reply: The revised manuscript contains a separate section about accuracy. The sentence has been removed.

Revised: P7-P8|8-3. Section 3.7 and 3.71

*p5l30: "... between output and targets." Be more specific. What are targets? Class assignments for manually selected pixels? Are feature vectors the same as targets?*

Reply: The revised manuscript contains a separate section about accuracy. Here the above mention sentences (P5|30) has been removed.

Revised: P8|5-16

*p5l33-34: meaning unclear: "It shows a trade-off between sensitivity ..." This sentence is a stub. Trade-off between sensitivity and what?*

Reply: The revised manuscript contains a separate section about accuracy. Here the above mention sentences (P5|33-34) have been removed.

Revised: P8|17-32

*p5l35: typo: prefect*

removed

*p6l6-9: What about unresolved porosity below the image resolution?*

Reply: If the XCT instrument is unable to resolve the porosity, the reconstruction algorithm may fail to transform the sinograms in to representative pixel values. Therefore, resulting stack of 2D slices in Cartesian coordinates will lack the corresponding pixel information and hence unsupervised and supervised techniques will not be able to resolve the porosity below image resolution.

Not revised

*Shouldn't the image-derived porosities be all much lower than the experimental porosity values, because they do not capture very small pores?*

Reply: Yes. The averaged porosity values obtained from XCT images of Andesite, Berea and Rotliegend Sandstone is lower compared to laboratory measurements. Figure 11 in page 19 has been revised accordingly.

*p6l10-15: Okay, so a higher fuzziness parameter shifts the pore/matrix threshold towards lower gray values, so that the volume fraction of pores is reduced? But why exactly is this the case?*

Reply: Say pore, mineral, matrix phases are three sets named

$C_i, C_j, C_k$

When, $FCM_{1.10} = \{C_i\}\{C_j\}\{C_k\}$ are segmented into disjoint sets.

Hence, volume fraction is comparable to K-means.

As $FCM_{1.10} \rightarrow FCM_{1.85} = \{C_i\}\{C_j\}\{C_{jx} \cup C_k\}$, where $C_{jx} = (C_j \cap C_k)$, i.e the pore phase remains

the same. The mineral phase reduces consequently increasing the matrix phase, i.e the histogram binning for $FCM_{1.85}$ is finer and as we select the distinct labels of the phases, the total volume fraction form the $C_k$ appears higher compared to the pore phase.

Not revised

*Also, in the conclusions (p8l16-18) you state that somehow FCM can distinguish between pores and pore-throats, which is not true, because it would mean that the algorithm could distinguish between different pore sizes or functional units of the pore space. All FCM does, is to evaluate the histogram (only grayscale information, no spatial information) and depending on how you set the fuzziness parameter, partial volume voxels are assigned to pores or matrix. Similar statement in p8l16-17.*

Reply: The assumption is that, pore and pore throat pixel values are slightly different to each other. Therefore, by relaxing or constraining the fuzzy parameter, the clustered class is either a disjoint or union set of pore and pore throat pixels.

Not revised

*p6l21-22: meaning unclear: "Morphological and filtering operations were performed based on the complexity of the segmented images". Which image was processed how?*

Reply: In the case of Rotliegend Sandstone (21 µm) as the XCT images were noisy we have used contrast filter to enhance the image. Whereas, for other XCT images (Berea, Andesite and Musli) as the resolution and contrast were sufficiently high (7.5 µm to 13 µm) using filters was not necessary.

Revised: The above information has added in the subsection image processing.

*p6l31: meaning unclear: "... and post processing of the unknown dataset." This is the first time you mention post processing. Do you mean with "post", that it is carried out after some tentative image segmentation is completed?*

Reply: What is meant by post processing is that, the gray scale pixel values of the (8 bit or 16 bit) raw images exposed to the classification model.

Not revised

*What do you do specifically during post-processing? Why is it somehow linked to the size of feature vectors?*

In the case of AANN

In the first step the classification model (Levenberg-Marquardt backpropagation method) is trained using segmented images obtained from K-means/FCM.

The number of slices (K-means/FCM segmented images) used for trained varied from 1-10.

In the second step based on the segmented class used to train the classification model. The raw (8 bits or 16 bit) image is scaled up to the range of this value. And thereafter exposed (or given) to the model to perform classification.

In the case of LS-SVM (Here, we take Synthetic ('Musli') as an example).

In the first step

Feature vector is a set of pixel values (row matrix). Each set (pixel values) represents a phase (pore, matrix, rock, cracks, trapped pores ect.) in the XCT image. These sets are disjoint sets.

Example: class 1 = A set containing 495 pixel values representing pores and microspores.

class 2 = A set containing 508 pixels values representing matrix.

class 3 = A set containing 192 pixels values representing boundaries that connect (or separate) microspores with pores and matrix. These can be seen in Figure 5 as bright stripes encapsulating different clusters of microspores.

class 4 = A set containing 332 pixels values representing mineral type 1.

class 5 = A set containing 35 pixels values representing mineral type 2.

class 6 = A set containing 73 pixels values representing noise in the image.

class 7 = A set containing 397 pixels values representing cracks and specks in the image.

LSSVM model is trained using the feature vectors and the class labels corresponding to these pixel values. The class label(s) is a row matrix of values from 1 to 7; same size as total number of representative pixels.

One that model is trained up to an optimal threshold.

In the second step, raw (8 bit or 16 bit) images are exposed to the trained model to perform classification.

We refer to the second step as post processing.

*p7l2: meaning unclear: "As a consequence, the individual (weak) classification models". What do you mean by weak?*

In general, weak learner or classifiers models are algorithms which perform classification with substantially high error rate − (< 0.5) but slightly better than random guessing. The main advantage to bootstrap such weak learner is to gain speed.

Revised: P6|14-16

*p7l5: meaning unclear: "... most appropriate class for each phase." Do you mean: ... for each pixel?*

Reply: Yes.

Revised: P10|21

*p7l11: meaning unclear: " ... cluster homogeneity is over-segmented ..." This makes no sense to me.*

Revised: P10|28-29

*p7l28-30: "As the hand-picked ... quality and speed". This sentence can probably only be understood by an absolute insider. Which material in which rock did you hand-pick to represent class four?*

Reply: please refer to the reply of p6l31 were we have addressed the post processing procedure.

Andesite ⇔ Class four: Matrix

Synthetic Sample ⇔ Class four: mineral phase

Rotliegend Sandstone ⇔ Class four: mineral phase

Berea Sandstone ⇔ Class four: mineral phase

*Why do you consider a mix of all phases and noise appropriate? For me this actually sounds like the method failed completely, when class four does not represent a single material.*

Reply: Please refer to the reply of p6l31. As mentioned above class four is a single material.

*p7l39: meaning unclear: "The initial growth of the leaf size ...".*

Revised: P11|16

*p8l1: typo: chucks*

Removed

*p8l1-3: meaning unclear: "Using a for-loop with an increment of from one to ten, ... ith fold."
Since you did not describe 10K cross validation in the first place, it is not possible to understand this sentence without background knowledge.*

Revised: P11|16-17

*p8l8-14: This paragraph should be part of the discussion and not the conclusions.*

Reply We disagree, we believe they are more appropriate in the conclusion section.

*p8l14: What is class six? Why is class six different from others?*

Reply: Feature vector is a set of pixel values (row matrix). Each set (pixel values) represents a phase (pore, matrix, rock, cracks, and trapped pores etc.) in the XCT image. These sets are disjoint sets.

class 6 = A set containing 73 pixels values representing noise in the image.

class 7 = A set containing 397 pixels values representing cracks and specks in the image.

Therefore, as we introduce noise, Ensemble classifiers algorithm seem to get confused. It is difficult to speculate why this happens. And, later stabilize itself on introduction of class 7.

Revised: P9|26-31

*p8l19-22: This conclusion is highly specific to the SOM method and cannot be generalized into something useful. You're basically saying the parameters that you've chosen a priori worked well. But maybe a hexagonal topology and a Euclidean distance function would have also done the job?*

Reply: During initial testing phase, different topologies were tested, like grid, hexagon and random topology in combination with Euclidian, cosine, Manhattan and box distance functions. Grid topology in combination with Manhattan distance yield slightly better result. Not revised.

*p8l26-30: Here in the conclusions you refer for the first time to "scaling". Why would you want to scale your images, when you're only interested in the segmentation results at the scale at which you acquired the image. Often a segmentation result at a coarse scale is of very little use. Therefore, the sentence "Additionally, the accuracy ..." is hard to follow.*

Reply: To achieve classification using back propagation feed forward neural networks. It is a recommended procedure that the training and testing data correlate to a certain extent. As we have used segmented data from unsupervised techniques to train the FFANN, the raw data set used for testing had to be scaled to minimize the error rate.

*p8l35-39: Of course the class labels should contain only one real phase in your rock, not only parts of it and not many simultaneously. Why is that an important conclusion?*

Reply: Our aim is to point out the best practice for accurate results.

Not revised

*Fig. 1: directly copied from older paper. It's not clear from the text, why this figure needs to be added.*

Revised – Figure 1. has been removed.

*Fig. 2: Why only show ten grayscale bins, when the original data is at least 8-bit, i.e. 256 bins?*

Reply: ten grayscale bins depict the best representation of different phases.

*Fig. 5: The legend suggest that there are also classes/colors with half labels, e.g. 0.5, 1.5, 2.5? Is this really the case?*

Reply: Indexing out the values for 0.5, 1.5, 2.5 showed empty matrix.

Revised: The legend has been revised

*Fig. 6: Figure caption is out of context.*

Revised

*Fig. 9: Fonts are too small. ROC curves are not self-explanatory. What does "-inf vs. 1" for Berea sandstone mean? What is probability of false alarm and probability of detection? How do you derive total accuracy from the individual curves?*

Revised: please refer to section 3.7.3. We don't know what does –inf values are, we speculate they can be random pixel values for crack and specks or negative values, which are well distinguished by the classification model (LS-SVM) from other classes labels.

---

## Author Comment (AC2) · 27 May 2016

Dear Editor:

We appreciate the efforts you and the reviewers have invested in our manuscript. We have addressed all relevant concerns of the reviewer and are submitting a revised version for your consideration.

We are aware, that the new content, with respect to the in both reviews mentioned paper of Chauhan et al., 2016, is only limited.

Notwithstanding of this, the paper we have submitted to SE is a completely new contribution with no redundancy to the previous one.

Also, we believe that it would be a nice contribution for this SI to round up things.

With respect to the technical comments we have revised the paper in such a way that the reviewer's comments are taken into account.

Following is an itemized list of reviewers' comments together with our response. Comments are reported in italic fonts (red) and our responses in regular (blue).

Sincerely,

On behalf of all authors Wolfram Rühaak

*Anonymous Referee #2*

*Review for: Swarup Chauhan et al. "Phase Segmentation of X-Ray Computer Tomography Rock Images using Machine Learning Techniques: an Accuracy and Performance Study"*

*Dear authors,*

*from my point of view your contribution is not significant enough to be published in Solid Earth. The main reason is the only marginal added value to your already published paper "Processing of rock core microtomography images: Using seven different machine learning algorithms" (Computer & Geosciences, 2016). Your statement in the abstract "Therefore, our investigation provides parameters that can help selecting the appropriate machine learning techniques for phase segmentation" is rather vague to support the idea to have another paper to your already published paper.*

See reply to reviewer #1

*Moreover, there are other very weak points in the paper:*

*- The introduction is too short and will give not a good overview of the topic. Do the authors will really not have a better overview?*

Revised: The introduction has been modified. P1-P3|29-4

*- Other segmentation techniques are not discussed in the manuscript. Maybe they are superior to the discussed seven methods.*

Reply: It is not clear what is meant with other segmentation techniques. The manuscript is not aimed towards the review of different segmentation techniques. The emphasis here is to show the capability of machine learning techniques to tackle phase segmentation problem. We propose ML techniques which can be used as one of the alternative to several other

segmentation techniques.

*- Nowadays it is a standard to apply several filters in the data processing workflow. They are not mentioned or discussed.*

Reply: In the case of Rotliegend Sandstone (21 µm) as the XCT images were noisy, contrast filter was used to enhance the image. Whereas, for other XCT images (Berea, Andesite and Musli) as the resolution and contrast were sufficiently high (7.5 µm to 13 µm) using filters did not show any noticeable change.

Revised: The above information has added in the subsection image processing.

*- The quality of the figures is bad. Partly they are too small too identify details (e.g. Figure 2, 3 or 9) or the labeling is not explained (what is FCM-1_35 in Figure 4 ???)*

Revised: The labeling and quality of the 2, 3 has been revised accordingly.

Reply: FCM was constrained at different membership function to check the segmentation quality. Hence, FCM-1.35 in figure 4 refers to the constrained membership value.

*In conclusion, the paper is not acceptable in its current form. There are simply too many things to correct.*

---

## Editor Comment (EC1) · S. Henkel (Editor) · 24 Jun 2016

Dear authors and referees, Thank you very much for your time and your detailed work which will increase the quality of the solid earth special issue "Pore-scale tomography & imaging – applications, techniques and recommended practice". I recognized both detailed reviews, comments and recommendations about the manuscript from: S. Chauhan, W. Rühaak, H. Anbergen, A. Kabdenov, M. Freise, T. Wille and I. Sass with the title: Phase Segmentation of X-Ray Computer Tomography Rock Images using Machine Learning Techniques: An Accuracy and Performance Study. The rigorous and reasonable revision of Dr. W. Rühaak and the well-executed responds to the reviews considerably increased the quality of the manuscript. The discussion with all editors of

this special issue thereafter led to the decision that this manuscript can be accepted after this revision. Additionally, it will have a positive scientific influence and will complete the scope of this volume.

Thank you all again for your contributions, Steven Henkel.